# PKCAM: Previous Knowledge Channel Attention Module

## Abstract

Attention mechanisms have been explored with CNNs, both across the spatial and channel dimensions. However, all the existing methods devote the attention modules to capture local interactions from the current feature map only, disregarded the valuable previous knowledge that is acquired by the earlier layers. This paper tackles the following question: Can one incorporate previous knowledge aggregation while learning channel attention more efficiently? To this end, we propose a Previous Knowledge Channel Attention Module(PKCAM), that captures channel-wise relations across different layers to model the global context. Our proposed module PKCAM is easily integrated into any feed-forward CNN architectures and trained in an end-to-end fashion with a negligible footprint due to its lightweight property. We validate our novel architecture through extensive experiments on image classification and object detection tasks with different backbones. Our experiments show consistent improvements in performances against their counterparts. We also conduct experiments that probe the robustness of the learned representations.

## 1 Introduction

Over the years, CNN architectures have evolved with many ideas to better deal with spatial image features. Moreover, their localized nature makes such features lack the global view of the image. Deeper architectures emerged that stack multiple convolution layers, known with different names; backbone, bottleneck, feature extractor, or encoder. The main feature of such architectures is their ability to cover spatial features at multiple scales. As we go deeper, the feature maps get smaller, while their content represents a wider region in the space, which gets us closer to better semantics of the image contents Luo et al. (2016). With the emergence of AlexNet Krizhevsky et al. (2012), many kinds of research investigate to further improve the performance of deep CNNs. Simonyan & Zisserman (2014) He et al. (2016) Szegedy et al. (2017) Szegedy et al. (2016) Srivastava et al. (2015) have sought to strengthen the CNNs by making it deeper and deeper as they have shown that increasing the depth of a network could significantly increase the quality of the learned representations. Many researchers are continuously investigating to further improve the performance of deep CNNs by incorporating attention mechanisms to exploit its ability to cover the relationship between the learned spatial features.

Attention modules, in general, are designed to suppress noise while keeping useful information by refining the learned features using attention scaling. By quoting from the human perception process Mnih et al. (2014) where the high-level information is used in guiding the bottom-up learning process by capturing more sophisticated features while disregarding irrelevant details. Human perception and visual attention Beck & Kastner (2009) Desimone (1998) Mnih et al. (2014) Desimone & Duncan (1995) is enhanced by top-down stimuli and non-relevant neurons will be suppressed in feedback loops. Referencing to human visual system, various different attention mechanisms Cao et al. (2019) Wang et al. (2018) Zhao et al. (2020) Vaswani et al. (2017) Wang et al. (2017) have been explored and integrated into deep CNNs. Attention mechanisms were introduced in the context of CNNs to capture the relations between features, either across the spatial dimension as in Carion et al. (2020) Hu et al. (2018a) ,or across channel-wise dimension as in Wang et al. (2018) Cao et al. (2019) Hu et al. (2018b) Lee et al. (2019) Wang et al. (2020) or across both dimensions as in Park et al. (2018) Woo et al. (2018) Fu et al. (2019) Wang et al. (2017) Linsley et al. (2018) Roy et al. (2018) Chen et al. (2017). Although these attention methods have achieved higher accuracy than their counterpart

Table 1: Comparison of existing attention modules in terms of whether previous knowledge cross-channel interaction (PKCCI), attention dimension, where C indicates channel attention and S indicates spatial attention, and lightweight or not.

| Methods | SE | ECA | SRM | GSop | GC | GE | CBAM | BAM | DAN | GALA | RAN | PKCAM |
|---|---|---|---|---|---|---|---|---|---|---|---|---|
| PKCCI | ✗ | ✗ | ✗ | ✗ | ✗ | ✗ | ✗ | ✗ | ✗ | ✗ | ✗ | ✓ |
| Attention | C | C | C | C | C | S | C + S | C + S | C + S | C + S | C + S | C |
| Lightweight | N/A | ✓ | ✓ | ✗ | ✗ | ✗ | ✗ | ✗ | ✗ | ✗ | ✗ | ✓ |

baselines which do not invoke any attention mechanisms in their architectures, they often bring higher model complexity and exploit only the current feature map while refining it, that's why we call it local attention mechanisms. Exploiting previous knowledge has been applied to image classification Huang et al. (2017) Iandola et al. (2014), image segmentation Ronneberger et al. (2015), tracking Ma et al. (2015), and human pose estimation Newell et al. (2016) where they obtain enhanced performance. DenseNets Huang et al. (2017) encourage feature reuse by connecting each layer to every other layer in a feed-forward fashion. U-Net Ronneberger et al. (2015) consists of two paths, which are contracting path to capture context and a symmetric expanding path that enables precise localization, where feature reuse is introduced through using skip connection between two paths. Driven by the significance of employing feature reuse while learning different tasks Szegedy et al. (2015) Huang et al. (2017) Iandola et al. (2014) Chen et al. (2016) Ma et al. (2015) Newell et al. (2016), a question arises: How can one incorporate previous knowledge aggregation while learning channel attention more efficiently?

To answer this question, we introduce PKCAM, a novel feature recalibration module based on channel attention, which improves the quality of the representations produced by a network using the global information to selectively emphasize informative features and suppress less useful ones. In contrast to the aforementioned attention mechanisms, our global context aware attention block obtains additional inputs from all preceding attention blocks, that have the same depth, and passes on its refined feature-maps to all subsequent blocks, creating global awareness from exploiting previous knowledge aggregation from earlier layers that can capture fine-grained information which is useful for precise localization while attending to features from earlier layers that can encode abstract semantic information, which is robust to target appearance changes.

The contributions of this paper are summarized as follows:

- We propose a simple and effective attention module, PKCAM, which can be integrated easily with any CNNs and applied across all it's blocks due to the lightweight computation of our novel architecture.
- We verify the effectiveness and robustness of PKCAM throughout extensive experiments with various baseline architectures on multiple tasks and datasets.
- Through detailed analysis along with ablation studies, we examine the internal behavior and validity of our method.

The rest of the paper is organized as follows, first, we discuss the related work, followed by the details of the proposed model. Then we present detailed ablation studies to settle on the best architectural design, and finally, illustrate the experimental setup for the various experiments we conducted for every contribution.

## 2 RELATED WORK

**Channel attention** Hu et al. (2018b) proposed SE block, squeeze, and excitation block, which comprises a lightweight gating mechanism that focuses on enhancing the representational power of the network by modeling channel-wise relationships using two fully connected layers. ECA-Net Wang et al. (2020) empirically shows avoiding dimensionality reduction in Hu et al. (2018b) by using a simple 1-D convolution layer, is important for learning channel attention and appropriate cross-channel interaction. SRM Lee et al. (2019) proposes a Style-based Recalibration Module, which adaptively recalibrates intermediate feature maps by exploiting their styles. Zhao et al. (2020) explore two variations of self-attention, pairwise and patchwise, that produce more powerful refined

Figure 1: Diagram of our Previous Knowledge Channel Attention Module (PKCAM). Given a $R$ aggregated features, PKCAM generates global aware channel weights by performing a fast 1D convolution of size $R$, accompanied by another 1D convolution, which represents global cross channel interaction, then fused with the standard local cross channel interaction.

features. The basic non-local block (NLB) Wang et al. (2018) aims at strengthening the features of the query position via aggregating information from other positions. GC-NetCao et al. (2019) introduces an abstract global context modeling framework, that could be summarized into two blocks: context modeling and transform block besides proposing a simplified local network as the context modeling and use SENet Hu et al. (2018b) as the transform block. GSoP Gao et al. (2019) obtain a covariance matrix by exploiting holistic image information using global second-order pooling, which is used for tensor scaling along channel dimension.

**Spatial attention**  GENet Hu et al. (2018a) consists of two operators that follow also the context modeling framework Cao et al. (2019), gather and excite operators. GENet Hu et al. (2018a) uses stridden depth-wise convolution which acts as the gather operator, which applies spatial filters to independent channels of the input, and a simple excite operator consists of sigmoid function and multiplication. Spatial Transformer Networks Jaderberg et al. (2015) tackle the lack of CNN ability to be spatially invariant to the input, by integrating a learnable module, the Spatial Transformer, which can be inserted into CNNs, giving neural networks the ability to actively spatially transform feature maps, conditional on the feature map itself. DETR Carion et al. (2020) stacks a spatial transformer after the CNN backbone to learn the interaction between each spatial position and its effect on different vision tasks, object detection, and instance segmentation.

**Spatial and channel attention**  BAM Park et al. (2018), CBAM Woo et al. (2018), DANet Fu et al. (2019), Residual attention network Wang et al. (2017), SCA-CNN Chen et al. (2017), scSE Roy et al. (2018) and GALA Linsley et al. (2018) show that taking the spatial axis into consideration besides channel axis boost the attention module accuracy. Given an intermediate feature map, they sequentially infer attention maps along two separate dimensions, channel and spatial, then the attention maps are multiplied to the input feature map for adaptive feature refinement.

Table 1 summarizes the existing attention modules in terms of whether previous knowledge cross-channel interaction, attention type where C means channel attention mechanism is used, S means spatial attention mechanism is used and C + S indicates that both attention mechanisms are used, and lightweight model.

## 3 METHODOLOGY

In this section, we first demonstrate an abstracted overview of our PKCAM. Then, we demonstrate the motivations to adopt the feature reuse concept via exploiting the previous knowledge to create a global aware attention block (i.e., PKCAM). Finally, dissecting our PKCAM by detailing it main blocks.

### 3.1 ABSTRACTED OVERVIEW OF OUR PKCAM

By scrutinizing the aforementioned channel attention techniques, as presented in Table 1, previous knowledge aggregation was not explored from the channel attention module perspective. Therefore we studied the previous knowledge cross-channel interaction by proposing PKCAM. The left part in Fig.**??** demonstrates an abstracted overview of our PKCAM, which exploits both local and global

feature maps while recalibrating the current feature map. PKCAM consists of two stacked modules: previous knowledge aggregation (PKA) and global cross channel interaction (GCCI). The PKA module covers the channel interactions across different preceding aggregated feature maps, while the GCCI module utilizes the refined features produce by the PKA module to model channel-wise relationships in a computationally efficient manner. GCCI modules could be any one of the on-the-shelf channel attention techniques which are studied in Section 2. Algorithm 1 demonstrates a pseudo algorithm for our PKCAM module to show how easily it could be implemented and integrated to any CNN architecture.

### 3.2 PREVIOUS KNOWLEDGE AGGREGATION BLOCK

In contradiction to the aforementioned channel attention techniques which relies on the current output of an arbitrary CNN block, our proposed PKCAM exploits both the current CNN block output , $x_0 \in \mathbb{R}^{H_0 \times W_0 \times C_0}$, and a range of earlier CNN blocks output , $X_p = [x_1, x_2, ..., x_R]$, where $R$ is the coverage region that delimits how many previous CNN blocks output will be consolidated along side the current CNN block, $x_1 \in \mathbb{R}^{H_1 \times W_1 \times C_1}$, $x_2 \in \mathbb{R}^{H_2 \times W_2 \times C_2}$, and $x_R \in \mathbb{R}^{H_R \times W_R \times C_R}$.

**Channel dimension alignment** In general, the earlier features $X_p$ have different channel dimensions, as the conventional is as we go deeper the depth is increased. Therefore, the first operation in our Previous Knowledge Aggregation(PKA) block is aligning the channel dimension among different CNN blocks. As $C_0 \geq C_1 \geq C_2 \geq C_R$, aligning operation can be

---

**Algorithm 1** PKCAM module algorithm
___
**Input**: X: Current feature map.
           P: List of preceding feature maps.
**Output**: $Z_1$: Learned channel scales
1: $R \leftarrow Length(Y)$
2: **for** r in range(R) **do**
3:     $P[r] \leftarrow SDA(P[r])$
4:     $P[r] \leftarrow CDA(P[r])$
5: **end for**
6: $S \leftarrow Stack(X, Y)$
7: $Y \leftarrow 1DConv(S, Kernel = R)$
8: $Z_1 \leftarrow 1DConv(Y, Kernel = 3)$
9: **return** $Z_1$

---

done be learnable upsampling techniques or a simple repeating operation to align with the channel dimension of the current CNN block $C_0$, producing channel aligned feature maps, $x'_0 \in \mathbb{R}^{H_0 \times W_0 \times C_0}$, $x'_1 \in \mathbb{R}^{H_1 \times W_1 \times C_0}$, $x'_2 \in \mathbb{R}^{H_2 \times W_2 \times C_0}$, and $x'_R \in \mathbb{R}^{H_R \times W_R \times C_0}$.

**Spatial dimensions alignment** Analogous to aligning the channel dimensions, the spatial dimensions; $H$ and $W$, is aligned through squeeze operation by adapting the general global average pooling equation as follows, $\widetilde{X} = \frac{1}{RWH} \sum_{k=1}^{R} \sum_{i=1}^{W} \sum_{j=1}^{H} X_{kij}$, where $\widetilde{X} \in \mathbb{R}^{R \times 1 \times 1 \times C_0}$ and represents the squeezed feature maps from the channel aligned aggregated feature maps $x'_i$, where $i = 0, 1, \ldots, R$, producing $\widetilde{X} = [\widetilde{x}_0, \widetilde{x}_1, ..., \widetilde{x}_R]$, $\widetilde{x}_0 \in \mathbb{R}^{1 \times 1 \times C_0}$, $\widetilde{x}_1 \in \mathbb{R}^{1 \times 1 \times C_0}$ and $\widetilde{x}_R \in \mathbb{R}^{1 \times 1 \times C_0}$.

**Previous knowledge cross-channel attention** Given the aggregated feature $\widetilde{X}$, previous knowledge cross-channel attention can be learned by $Y = f(\widetilde{X})$, where $Y \in \mathbb{R}^{1 \times 1 \times C_0}$ , $f(\widetilde{X}) = W'\widetilde{X}$, and $W'$ could take one of the following forms,

$$
W' = \begin{cases} W'_1 = \begin{bmatrix} W'_{1,1} & \cdots & W_{1,RC_0} \\ \vdots & \ddots & \vdots \\ W'_{RC_0,1} & \cdots & W'_{RC_0,RC_0} \end{bmatrix} \\ W'_2 = \begin{bmatrix} W'_{1,1} & 0 & \cdots & 0 \\ 0 & W'_{2,2} & \cdots & 0 \\ \vdots & \vdots & \ddots & \vdots \\ 0 & 0 & \cdots & W'_{RC_0,RC_0} \end{bmatrix} \end{cases} \tag{1}
$$

where $W'_1$ is a $RC_0 \times RC_0$ parameter matrix which learns previous knowledge interaction in conjunction with cross-channel interaction. In contrast $W'_2$ is a $1 \times RC_0$ parameter matrix which learns previous knowledge interaction and channel interaction neglecting the cross channel relations. The key difference between $W'_1$ and $W'_2$ is that $W'_1$ considers previous knowledge cross-channel

Table 2: Comparison of different previous knowledge Aggregation(PKA) techniques on the Tiny-ImageNet dataset. Where 1-D Conv., Sum and FC stands for one dimensional convolution layer Eq.3, summation Eq.2, and fully connected layer Eq.1 respectively.

| PKA | 1-D Conv. | Sum | FC | 1-D Conv. | Sum | FC | 1-D Conv. | Sum | FC |
|---|---|---|---|---|---|---|---|---|---|
| | ResNet-18 | | | ResNet-34 | | | ResNet-50 | | |
| Acc. | 55.70 | 55.28 | 54.63 | **56.94** | 56.26 | 56.52 | **57.89** | 56.18 | 56.41 |
| #.P (M) | **10.749** | 10.749 | 11.413 | **20.389** | 20.389 | 23.049 | **22.824** | 22.824 | 65.387 |
| GFLOPs | **2.075** | 2.075 | 2.076 | **4.329** | 4.329 | 4.331 | **4.878** | 4.878 | 4.878 |

interaction while $W_2'$ does not, leading $W_1'$ to be more complex than $W_2'$. Interpreting Eq. 1 to neural networks $W_1'$ and $W_2'$ can be regarded as a fully connected layer and depth-wise separable convolution layer respectively. However, obviously from Eq. 1, $W_1'$ and $W_2'$ have a tremendous number of parameters, driving to high model complexity, especially for large channel numbers as mainly $C_0 >> R$.

Therefore, we divide learning the previous knowledge cross-channel interaction into two sub-modules as shown in Fig. 1, learning previous knowledge interaction, and exploiting the cross-channel interaction. Consequently, in contrast to Eq. 1, $f(\widetilde{X})$ is splitted into two cascaded functions, $f_1(\widetilde{X})$ and $f_2(\widetilde{X})$, where $f_1(\widetilde{X})$ is responsible to learn the previous knowledge channel interaction and $f_2(\widetilde{X})$ is responsible to learn the cross-channel interaction.

**Previous knowledge channel interaction**   Previous knowledge channel attention can be learned by Eq. 2, where for each channel the global information is aggregated using simple summation operation, where no learnable parameters are invoked.

$$Y = f_1(\widetilde{X}) = \sum_{L=1}^{C_0} \sum_{K=1}^{R} \widetilde{X}_{LK} \tag{2}$$

A possible compromise between Eq. 1 and Eq. 2 is Eq. 3, where a tiny number of parameters are used whereas $W' \in \mathbb{R}^{1 \times 1 \times R}$ compared to the tremendous number of parameters that are invoked in Eq. 1 while learning the previous knowledge channel interaction. From the perspective of the convolution neural network, Eq. 3 could be readily interpreted to a 1-D convolution layer with kernel $k = \widetilde{W}$.

$$Y = f_1(\widetilde{X}) = \sum_{L=1}^{C_0} W' \widetilde{X}_L \tag{3}$$

**Global cross-channel interaction**   Global cross-channel interaction could be learned by adopting one of the local channel attention modules Hu et al. (2018b) Cao et al. (2019) Wang et al. (2018) Lee et al. (2019) Wang et al. (2020) Gao et al. (2019) producing $Z_1 = f(Y)$ where $Z_1 \in \mathbb{R}^{1 \times 1 \times C_0}$. Wang et al. (2020) Gao et al. (2019) Hu et al. (2018b) Cao et al. (2019) Wang et al. (2018) Lee et al. (2019) refer to the term global as they are taking into consideration the whole spatial dimension from the fed features using GAP - Global Average Pooling. In contrast we refer to the term global as previous knowledge aggregation.

## 4 EXPERIMENTS

In this section, we perform controlled ablation experiments to settle on the best design for our proposed module and assess its sub-modules. Then we evaluate the performance of the proposed Previous Knowledge Attention module, on a series of benchmark datasets across different tasks including Tiny-ImageNet Le & Yang (2015), i.e., discussed in the appendix, and ImageNet Deng et al. (2009) for the classification task, and KITTI Geiger et al. (2012) for detection. Finally, We conduct empirical experiments that probe the robustness of the representations learned by PKCAM, compared to convolutional baselines and other attention mechanisms.

Table 3: Comparison of the various basic attention modules in our proposed module PKCAM.

| Basic Attention Module | SE | SRM | ECA | SE | SRM | ECA |
|---|---|---|---|---|---|---|
| | Tiny-ImageNet | | | ImageNet | | |
| Resnet-18 | 54.07 | 55.10 | **55.70** | 70.65 | 70.75 | **70.83** |
| Resnet-34 | 56.61 | 56.52 | **56.94** | 73.98 | **74.48** | 74.39 |
| Resnet-50 | 57.42 | 57.12 | **57.89** | 76.86 | 77.42 | **77.56** |

## 4.1 IMPLEMENTATION DETAILS

For classification task two datasets are used, i.e., Tiny-ImageNet dataset Le & Yang (2015) and ImageNet dataset Deng et al. (2009), to evaluate our proposed module and show its effectiveness, where the same data augmentation and hyper-parameter settings in Hu et al. (2018b) are adopted. For the Tiny-ImageNet dataset Le & Yang (2015), input images are randomly cropped to $64 \times 64$ with random horizontal flipping. For the ImageNet dataset Deng et al. (2009), a $224 \times 224$ crop is randomly sampled from an image or its horizontal flip, with the per-pixel RGB mean value subtracted. All models are trained for 100 epochs from scratch, using the weight initialization strategy described in He et al. (2015) and the initial learning rate is set to 0.1 and decreased by a factor of 10 every 30 epochs. Stochastic gradient descent (SGD) with weight decay of $1e-4$, the momentum of 0.9, and mini-batch size of 32 is used for Tiny-ImageNet Le & Yang (2015), and 256 for ImageNet Deng et al. (2009). Our module is implemented in Python using the PyTorch framework using four PCs with Intel Xeon(R) 4108 1.8GHz CPU, 64G RAM, Nvidia Titan-XP.

## 4.2 ABLATION STUDIES FOR INTERNAL DESIGN

We have conducted two to settle on the best internal design and analyze the effectiveness of each component in our PKCAM. As shown in Fig. 1, our proposed module consists of two cascaded blocks, i.e., previous knowledge aggregation block (PKA) and global cross channel interaction block (GCCI). Therefore two ablations studies are conducted to settle on the best internal design and analyze the effectiveness of each one of them. The first ablation investigating the different approaches for previous knowledge channel interaction that are discussed in Section 3.2. Then, we assess the choice of basic attention modules that are used in the global cross channel interaction modules as shown in Fig. 1.

### 4.2.1 PREVIOUS KNOWLEDGE AGGREGATION

We have investigated empirically the different global channel interaction techniques that were described in detail in Section3.2. As shown in Table8 exploiting the previous knowledge channel interaction through 1-D Conv. layer by following Eq.3 is the best compromise to cope with the trade-off between performance and complexity, where it shares almost the same model complexity (i.e., network parameters and FLOPs) with the original ResNet while at the same time it boosts the accuracy. Based on the aforementioned results in Table 8, our novel approach follows the compromised combination while exploiting the previous knowledge cross channel interaction by following Eq.3 to capture the previous knowledge channel interaction.

### 4.2.2 GLOBAL CROSS CHANNEL INTERACTION

We next assess the choice of basic attention modules that are used in the global cross channel interaction module as shown in Fig.1. Three channel attention mechanisms are evaluated on the Tiny-Imagenet dataset Le & Yang (2015) and ImageNet dataset, including SE-Net Hu et al. (2018b), SRM Lee et al. (2019), and ECA-Net Wang et al. (2020). As shown in Table 3 ECA-Net achieves the best accuracy across different ResNet backbones. However, building up our PKCAM using other channel attention mechanism boost the accuracy as shown in Table 3 compared to their original results that are mentioned in Table 9 and Table 5.

Table 4: Comparison of the various ways to integrate PKCAM into CNNs using the ImageNet dataset.

| Integration Methodology | All blocks | | Last block | | LCAM+PKCAM | | PKCAM | |
|---|---|---|---|---|---|---|---|---|
| | Top-1 | FPS | Top-1 | FPS | Top-1 | FPS | Top-1 | FPS |
| Resnet-18 | 70.78 | 112 | **70.78** | **138** | 70.78 | 138 | **70.83** | **160** |
| Resnet-34 | 74.29 | 69 | **74.36** | **83** | 74.36 | 83 | **74.39** | **93** |
| Resnet-50 | 77.19 | 42 | **77.22** | **69** | 77.22 | 69 | **77.56** | **74** |

Table 5: Comparisons with state-of-the-art attention modules on ImageNet in terms of the number of parameters (#P.) in millions, GFLOPs, top-1, and top-5 accuracy. Top-1 relative improvement results are reported between parentheses w.r.t SENet improvement over Vanilla Resnet.

| Methods | #.P.(M) | GFLOPs | FPS | Top-1 | Top-5 | #.P.(M) | GFLOPs | FPS | Top-1 | Top-5 | #.P.(M) | GFLOPs | FPS | Top-1 | Top-5 |
|---|---|---|---|---|---|---|---|---|---|---|---|---|---|---|---|
| | ResNet-18 | | | | | ResNet-34 | | | | | ResNet-50 | | | | |
| ResNet | 11.14 | 1.699 | 247 | 70.42 | 89.45 | 20.78 | 3.427 | 139 | 73.31 | 91.40 | 24.37 | 3.86 | 110 | 75.2 | 92.52 |
| +SENet | 11.23 | 1.700 | 146 | 70.59 | 89.78 | 20.93 | 3.428 | 79 | 73.87 | 91.65 | 26.77 | 3.87 | 54 | 76.71 | 93.38 |
| +CBAM | 11.23 | 1.700 | 92 | 70.73(182%) | 89.91 | 20.94 | 3.428 | 48 | 74.01(125%) | 91.76 | 26.77 | 3.87 | 36 | 77.34(141%) | 93.69 |
| +ECANet | 11.14 | 1.699 | 179 | 70.75(194%) | 89.74 | 20.78 | 3.427 | 100 | 74.13(146%) | 91.68 | 24.37 | 3.86 | 56 | 77.39(145%) | 93.60 |
| +PKCAM | **11.14** | **1.699** | 160 | **70.83**(241%) | **89.96** | **20.78** | **3.427** | 93 | **74.39**(192%) | **91.81** | 24.37 | 3.86 | 74 | **77.56**(156%) | **93.70** |

## 4.3 ABLATION STUDIES FOR INTEGRATING PKCAM

### 4.3.1 DO WE NEED TO INJECT PKCAM AT EACH BLOCK?

The majority of CNN backbones have the same convention in their structure that consists of cascaded blocks that form a stage, then the basic structure of the stage is repeated with some modifications to form the backbone architecture. Driven by this, we empirically demonstrate that, placing our PKCAM in the last block of each stage only is sufficient to boost the accuracy by learning more powerful representations, and there is no need to inject our module after each block to re-calibrate its channels. Fig. **??** demonstrates the two different approaches. Injecting PKCAM at the last block will be sufficient as it aggregate the whole previous knowledge within the same stage, exploiting this feature enables us to overcome all local attention modules in terms of model speed and complexity. Table 4 shows that injecting PKCAM at the last block only superior injecting it in the whole blocks in terms of Top-1 accuracy and the inference time.

### 4.3.2 PKCAM VS. LCAM

Local channel attention modules (LCAM) showed their ability in learning powerful representations, however exploiting the previous knowledge reinforcements the learned scales to be more representative. Table 4 demonstrates that our module is powerful enough to totally replace the known LCAM, that are discussed at Sec. 2, instead of leveraging both of them. Upper part in Fig. **??** shows the fusion approach between LCAM and PKCAM, while the right part in Fig.1 shows using PKCAM only while integrating it to an arbitrary CNN.

## 4.4 IMAGE CLASSIFICATION

In this section, we evaluate the performance of proposed PKCAM network on classification benchmark datasets including ImageNet Deng et al. (2009) and Tiny-ImageNet Le & Yang (2015), that is mentioned in the appendix. All the classification experiments follows the same training procedure that is discussed in Sec. 4.1. The evaluation metrics incorporate both efficiencies (i.e., network parameters (#P.) in millions, inference frame rate per second (FPS), and floating-point operations per second (FLOPs) in Gigas, and effectiveness (i.e., Top-1 accuracy).

ImageNet LSVRC 2012 dataset Deng et al. (2009), which contains $10^3$ classes with 1.2 million training images, $50 \times 10^3$ validation images, and $100 \times 10^3$ test images. The evaluation is measured on the non-blacklist images of the ImageNet LSVRC 2012 validation set.

We compare our PKCAM module with several state-of-the-art attention methods using ResNet family. Efficiency and effectiveness are measured, and the results are reported in Table 5 from their original papers besides reproducing ECA-Net results as we notice there are difficulties in reproducing and

Table 6: Zero-shot testing: Analyzing the robustness of trained networks on Tiny-ImageNet.

| Robustness | Vanilla | SE | ECA | PKCAM | Vanilla | SE | ECA | PKCAM | Vanilla | SE | ECA | PKCAM |
|---|---|---|---|---|---|---|---|---|---|---|---|---|
| | ResNet-18 | | | | ResNet-34 | | | | ResNet-50 | | | |
| No-Rotation | 53.33 | 53.71 | 53.76 | **55.70** | 55.90 | 56.08 | 55.66 | **56.94** | 56.11 | 57.78 | 56.59 | **57.89** |
| 90° | 20.56 | 20.60 | 21.80 | **21.83** | 21.68 | 21.75 | 22.70 | **22.71** | 23.04 | 25.42 | 25.53 | **25.68** |
| 180° | 25.20 | 25.85 | 26.59 | **27.18** | 26.60 | 26.42 | 27.57 | **27.57** | 28.88 | 32.40 | 31.74 | **32.44** |
| 270° | 20.71 | 20.98 | 21.49 | **22.26** | 21.47 | 21.74 | 22.48 | **23.59** | 22.85 | 25.69 | 25.85 | **26.04** |

Table 7: Comparisons with state-of-the-art attention modules on KITTI-RGB in terms of mAP using YOLOV3 on Resnet-18 and 34 backbones.

| | Vanilla | SE | ECA | CBAM | BAM | SRM | PKCAM |
|---|---|---|---|---|---|---|---|
| R-18 | 57.87 | 59.32 | 58.55 | 57.90 | 59.61 | 59.20 | **59.66** |
| R-50 | 64.19 | 65.08 | 64.34 | 64.18 | 65.10 | 64.82 | **65.21** |

verifying their results[1]. We adopt the same training setup as He et al. (2016) Hu et al. (2018b) for fair comparison as discussed at Section 4.4. Results show that our proposed PKCAM achieves the best accuracy besides be the lightest model compared to other attention modules. Top-1 relative improvement results is reported between parentheses in green w.r.t SENet improvement over Vanilla Resnet.

## 4.5 OBJECT DETECTION

We evaluate our proposed PKCAM on object detection task using Faster R-CNN Ren et al. (2015) on the MS COCO dataset Lin et al. (2014) and using YOLOV3 Redmon & Farhadi (2018) on the KITTI-RGB dataset. KITTI-RGB Geiger et al. (2012) consists of 7,481 training images and 7,518 test images, comprising a total of 80,256 labeled objects of eight different classes. Each image has 3 RGB color channels and pixel dimensions $1242 \times 375$ which is resized to $224 \times 224$. We follow the same training setup as mentioned at Section 4.4. As shown in Table 7, PKCAM considerably improves the accuracy more than other attention modules compared to the baseline He et al. (2016). The MMDetection Chen et al. (2019) framework is used to guarantee fair comparison between different channel attention mechanisms

## 4.6 ROBUSTNESS

We have conducted experiments to probe the robustness of the representations learned by our proposed module PKCAM, compared to other channel attention mechanisms on Tiny-ImageNet Le & Yang (2015), where the testing images are rotated deliberately in one of three ways: clockwise 90°, clockwise 180°, clockwise 270°, where these rotations were not scrutinized at the training. As shown in Table 6, PKCAM is less vulnerable than other attention modules.

## 5 DISCUSSION

### 5.1 TOP-1 VALIDATION ACCURACY CURVES

To provide some insight into influence of our PKCAM module on the optimisation of these models, example training curves for runs of the baseline architectures and their respective PKCAM counterparts are depicted in Fig. 2. We observe that PKCAM blocks yield a steady improvement throughout the optimisation procedure. Moreover, this trend is fairly consistent across a range of network architectures considered as baselines.

### 5.2 DISSECTING THE PRODUCED LEARNABLE SCALES

To further analyze the effect of our PKCAM module on learning channel attention, we visualize the scales learned by our novel PKCAM modules and compare it against ECA module. For this

---

[1]Referring to issues number 21, 52, 62, 24, 46, and 58 from the official ECA-Net implementation.

Table 8: Comparisons with state-of-the-art attention modules on the MS-COCO dataset using Faster R-CNN detector based on Resnet-50 backbones.

| Method | #.P.(M) | GFLOPs | FPS | AP | $AP_{50}$ | $AP_{75}$ | $AP_S$ | $AP_M$ | $AP_L$ |
|---|---|---|---|---|---|---|---|---|---|
| ResNet-50 | 41.53 | 207.07 | 12 | 36.4 | 58.2 | 39.2 | 21.8 | 40.0 | 46.2 |
| +SENet | 44.02 | 207.18 | 7 | 37.7 | 60.1 | 40.9 | 22.9 | 41.9 | 48.2 |
| +ECANet | 41.53 | 207.18 | 8 | 38.0 | 60.6 | 40.9 | 23.4 | 42.1 | 48.0 |
| +PKCAM | **41.53** | **207.18** | **10** | **38.3** | **60.9** | **41.0** | **23.9** | **42.4** | **48.2** |

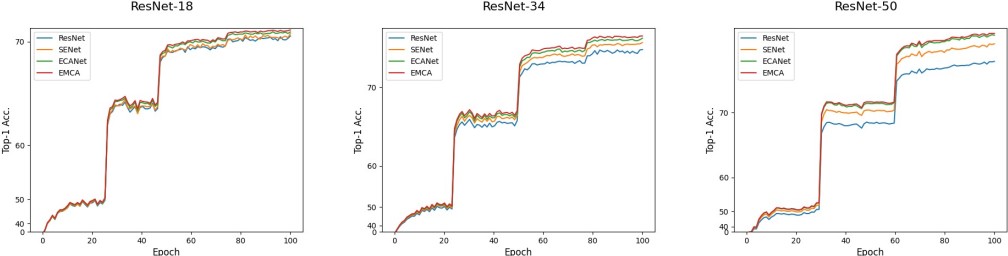

Figure 2: Training ResNet, and local channel attention modules (LCA) baseline architectures and their PKCAM counterparts on ImageNet validation set. PKCAM exhibit improved optimisation characteristics and produce consistent gains in performance which are sustained throughout the training process.

experiment, we adopt ResNet-18 as backbone models, and illustrate scales of first block only for each stage as discussed in Sec. 4.2 in the main script. In contrast to ECA setup, where a random sample consists of four classes only from ImageNet dataset are used, which are hammerhead shark, ambulance, medicine chest and butternut squash, we have used a more generic and fair way to analyze the learned scales by averaging them over the whole validation dataset instead of using only four selected classes. Fig. 3 visualizes the channel learned scales for each first block from each stage for both modules; PKCAM in blue and ECA in orange.

As shown in Fig. 3,ECA scales have larger variance than our learned scales, where ECA module's authors claim that it indicates a better discriminative ability which necessarily reflect the quality of the learned scales. Accordingly, a question is arising, is our main focus is to learn a discriminative scales, or to learn a more representative ones? We argue the claim that says the more discriminative scales indicates a more useful ones. As the most important a more representative scales not discriminative ones. This could be shown as all the learned channels is important and we can't exclude any one of them unlike the learned ECA scales which indicates that we can totally prune some of the learned scales, this shows deficiency in the learning process where not the full network capacity was exploited. Our scales are shown empirically that they are more representative as they boost the accuracy in a consistent manner over different architectures and different tasks.

## 6 CONCLUSION

In this paper, we concentrate on determining an effective channel attention module with low model complexity. To this end, we propose efficient channel attention (PKCAM). Because of the lightweight

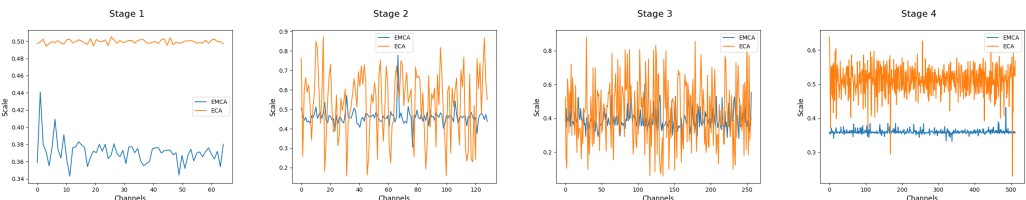

Figure 3: Comparison for the learned channel scales by our novel PKCAM module against ECA modules. Better view with zooming in.

computation of the PKCAM block, it can be integrated into all modern CNN architectures across the whole layers and trained end-to-end. While most previous works utilized uni-scale features, PKCAM is designed to employ the ability of global information while recalibrating feature maps. Our experiments demonstrate that simply inserting PKCAM into standard CNN architectures boosts the performance across different tasks. Furthermore, we verify the robustness of the representations learned by PKCAM and its generalization ability via zero-shot experiments to rotated images.

### AUTHOR CONTRIBUTIONS

If you'd like to, you may include a section for author contributions as is done in many journals. This is optional and at the discretion of the authors.

### ACKNOWLEDGMENTS

Use unnumbered third level headings for the acknowledgments. All acknowledgments, including those to funding agencies, go at the end of the paper.

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

## A  APPENDIX

In this section we will demonstrate more experiments on Tiny-ImageNet dataset.

Tiny-ImageNet Le & Yang (2015) that is also called Micro-ImageNet, is a classification challenge that is similar to the full ImageNet ILSVRC challenge. Micro-ImageNet comprises 200 classes. Each class has 500 images for training. The test set includes 10,000 images. A $64 \times 64$ crop is randomly sampled from an image or its horizontal flip, with the per-pixel RGB mean value subtracted.

Table 9: Comparisons with state-of-the-art attention modules on Tiny-ImageNet in terms of the number of parameters (#P.) in millions, GFLOPs, FPS, and top-1 accuracy.

| Methods | #.P (M) | GFLOPs | FPS | Top-1 | #.P (M) | GFLOPs | FPS | Top-1 | #.P (M) | GFLOPs | FPS | Top-1 |
|---|---|---|---|---|---|---|---|---|---|---|---|---|
| | ResNet-18 | | | | ResNet-34 | | | | ResNet-50 | | | |
| Vanilla | 10.749 | 2.074 | 352 | 53.33 | 20.389 | 4.327 | 195 | 55.90 | 22.802 | 4.861 | 150 | 56.11 |
| SE | 10.832 | 2.075 | 202 | 53.71 | 20.539 | 4.329 | 131 | 56.50 | 25.201 | 4.870 | 95 | 57.78 |
| ECA | 10.749 | 2.075 | **242** | 53.76 | 20.389 | 4.329 | 143 | 55.66 | 22.803 | 4.868 | 96 | 56.59 |
| SRM | 10.753 | **2.074** | 168 | 53.39 | 20.396 | **4.327** | 107 | 55.34 | 22.831 | **4.861** | 92 | 57.02 |
| CBAM | 10.835 | 2.075 | 132 | 55.06 | 20.544 | 4.329 | 68 | 55.27 | 25.218 | 4.867 | 49 | 56.79 |
| BAM | 10.771 | 2.080 | 201 | 55.00 | 20.411 | 4.333 | **145** | 55.97 | 23.144 | 4.962 | **119** | **58.37** |
| PKCAM | **10.749** | 2.075 | 215 | **55.70** | **20.389** | 4.329 | 136 | **56.94** | **22.803** | 4.868 | 101 | 57.89 |

We compare our PKCAM module with several state-of-the-art attention methods using three variants of ResNet backbone He et al. (2016) which are ResNet-18, ResNet-34, and ResNet-50 on the Tiny-Imagenet dataset Le & Yang (2015), including SENetHu et al. (2018b), ECANetWang et al. (2020), SRM Lee et al. (2019), CBAMWoo et al. (2018), and BAMPark et al. (2018). For a fair comparison, we ran all experiments by following the same setup that was mentioned above. As shown in Table 9, our PKCAM module shares almost the same model complexity with the original ResNets variants He et al. (2016), ResNet-18, ResNet-34, and ResNet-50, while achieving 2%, 0.7%, and 1.78% gains in Top-1 accuracy, respectively. Comparing with state-of-the-art counterparts (i.e., SENetHu et al. (2018b), ECANetWang et al. (2020), SRM Lee et al. (2019), CBAMWoo et al. (2018), and BAMPark et al. (2018)), PKCAM obtains better or competitive results while availing lower model complexity. Inference time is measured on a PC equipped with TITAN Xp GPU and an Intel(R) Xeon Silver 4112 CPU@2.60GHz.

Please note that, Our code will be published once the paper is accepted, also it is attached in the supplementary material compressed file.

