# OpenReview forum: "PKCAM: Previous Knowledge Channel Attention Module"
_ICLR.cc/2022/Conference — ICLR 2022 Submitted_

### Official Review · Reviewer_HH21 · 2021-10-31

**Correctness:** 2
**Technical Novelty And Significance:** 2
**Empirical Novelty And Significance:** 2
**Recommendation:** 3
**Confidence:** 4

**Main Review:**

This paper is not well prepared for review with many incomplete and inconsistent expressions, e.g., the Fig.?? in Pages 3 and 7, the EMCA in Figure 2 and Figure 3 but the PKCAM in the captions, the CDA and SDA in Figure 1 without any explanation in the caption and text, the first SDA then CDA operation in both Figure 1 and Algorithm 1 but inverse order in Section 3.2, etc.

The key contribution of this work is to consider aggregating the previous feature maps from earlier CNN blocks, but this idea has already been proposed for CNNs such as the DenseNet. Besides, the implementation of PKCAM is not special and actually the commonly used spatial and channel attention. Thus the paper has limited technical novelty.

For the ablation study, most of the analysis is actually related to the current techniques used in attention design (see Table 2). Besides, most of the comparison results show that the PKCAM is not superior to the compared counterparts a lot (refer to Tables 4, 5, 7, and 8 as well as Figure 2). Therefore, the advantage of PKCAM is not obvious, that is, this work is not convincing empirically.

From Figure 3, it can be seen that the PKCAM has a big difference with ECA, however, the quantitative comparison shows very small difference between PKCAM and ECANet, while the explanation of the last paragraph in Section 5 is not reasonable, especially how to evaluate the discriminative and representative ability of an attention module.

The paper slightly exceeds the paper limit of ICLR, and the writing needs to be carefully revised for typos and grammar errors.

**Summary Of The Paper:**

This paper introduces an attention module, namely PKCAM for Previous Knowledge Channel Attention Module, which can be integrated into CNNs. The proposed PKCAM mainly aggregates the feature maps of earlier CNN blocks for aggregating previous knowledge. The ablation study is conducted for validating the effect of PKCAM in CNNs as well as its different design. Image classification and object detection experiments are considered for demonstrating the effectiveness of PKCAM in different ResNet architectures.

**Summary Of The Review:**

The paper is not ready for review, also the contribution and analysis are limited. Thus, I vote for rejecting.

---

### Official Review · Reviewer_BFC2 · 2021-10-31

**Correctness:** 3
**Technical Novelty And Significance:** 2
**Empirical Novelty And Significance:** 2
**Recommendation:** 5
**Confidence:** 4

**Main Review:**

This paper proposes an attention module with the knowledge from the previous layers also included. I have two concerns about this idea.

Firstly, the idea to use the previous knowledge is not new. In the works of self-knowledge distillation, they use the previous layers knowledge to do the distillation (i.e. Be Your Own Teacher: Improve the Performance of Convolutional Neural Networks via Self Distillation). So, the idea of utilizing previous knowledge is not new and thus the contribution, for me, is not enough.

Secondly, the authors use the previous knowledge to build a better attention module. This idea is too intuitive. I don’t think the attention module has long-term relationship.


Some small typos in writing:
- The last line of page 3, there is something wrong with the figure (actually, all the citations for figures are wrong)
- The second row of sec 3.2, which rely on…


**Summary Of The Paper:**

This paper explores a new kind of attention mechanism, which can be easily integrated into any cnn models.

The proposed method is called previous knowledge channel attention module, whose aim is to make use of the knowledge from the previous layers.

The methods are validated on the image classification and object detection tasks.


**Summary Of The Review:**

-	What is the main contribution or motivation of this paper. If it is only about using the previous knowledge, it is not creative enough.
-	Try to explain or proof the working mechanism of the proposed PKCAM, which should be made more solid.
-	Try to polish the paper.

---

### Official Review · Reviewer_2Qrr · 2021-11-01

**Correctness:** 2
**Technical Novelty And Significance:** 2
**Empirical Novelty And Significance:** 2
**Recommendation:** 3
**Confidence:** 5

**Main Review:**

Strengths:
1. The previous feature maps are used to compute channel attention for the current feature map, showing some differences with existing methods. Besides, the proposed PKCAM is only applied to the end of the last stages, leading small parameters and FLOPs.
2.	The proposed method seems easy to implement.

Weaknesses:
1.	The technical contribution of this work seems limited. The core of PKCAM is to exploit feature maps from previous layers for computing channel attention of the current feature map. For me, it is still not very clear why previous feature maps can help to compute channel attention of the current feature map. In other word, what are the merits brought by the previous feature maps? Furthermore, the core components of PKCAM (i.e., SDA and CDA) show similar with existing works, e.g., SE and ECA blocks.
2.	The experimental results are not very convincing. Specifically,
(1)	The improvement of PKCAM over existing methods is very marginal in terms of accuracy and model complexity, which hardly verify the effectiveness of previous knowledge.
(2)	Experiments could be further strengthened. More stronger backbones (e.g., ResNet-101 for classification and Faster R-CNN/ Mask R-CNN for object detection) could be used to further evaluate performance of the proposed method.
(3)	Some experimental results need further discussions. For example, why FPS of ECANet increases two times from ResNet-34 to ResNet-50, while others only increase less 1.5 times? Why LCAM+PKCAM is inferior to PKCAM? How to combine LCAM with PKCAM? What are the details of LCAM?
(4)	Why PKCAM is only applied to the end of each stage? Why PKCAM only inserted into the last stage achieves the best results?
3.	The writing could be further polished. Specifically,
(1)	In Section 3.2, the details on how to select the previous knowledge seem missing, i.e., How to select R feature maps? and How about effect of number of R?
(2)	Which method is used for channel dimension alignment?
(3)	The orders of channel dimension alignment and spatial dimension alignment described in Eqn.(2) (from 1 to R) are inconsistent with those in Figure 1 (from 0 to R).
(4)	The dimension of fully-connected layer in Eqn.(1) seems confusing.
(5)	The sum operations in Eqn.(2) and Eqn.(3) are confusing.
(6)	Many citations are missing, e.g., Figure ???


**Summary Of The Paper:**

This paper computes channel attention by considering feature maps across different layers, namely previous knowledge channel attention module (PKCAM).  The developed PKCAM is achieved by two steps: (1) the previous knowledge aggravation module is used to aggregate channel information of multiple feature images; and (2) the global cross channel interaction module models the correlation among channels to compute attention weights. The experiments are conducted on image classification and object detection tasks.

**Summary Of The Review:**

In my opinion, the technical contribution of this work seems limited, while experiments and writing could be further polished.

---

### Official Review · Reviewer_cAgA · 2021-11-02

**Correctness:** 2
**Technical Novelty And Significance:** 2
**Empirical Novelty And Significance:** 2
**Recommendation:** 3
**Confidence:** 4

**Main Review:**

I think the descriptions about the core PKCAM module are unclear and seem to be incorrect. As shown on Page 4,  Y=W\widetilde(X) and thus Y should be a feature vector with the dimension of RC_0. However, the authors claim that Y is a feature vector with dimension of C_0. Do I miss something? Besides, the authors state to split the function f() into two functions f_1() and f_2(). However, I can not find the definition of f_2().

The authors split the PKCAN module into a summation followed by a linear mapping. It can indeed reduce the complexity. But why such simplicity works? More analysis should be provided.

The authors claim that the module can be easily integrated into different network architectures. However, they merely conduct experiments with the Reset series as baselines. I think the experiments with other architecture s such as mobile net should also be provided. Or the experiments can not support the statements.

More recent works [1, 2] also adopt channel and spatial attention for feature enhancement. These works should also be included for analysis and comparisons.

I think the work is somewhat a draft for the current version. There are many reference errors throughout the paper, e.g., fig. ?? on Page 3 and fig. ?? on Page 7.
[1] Hou et al., Coordinate Attention for Efficient Mobile Network Design Coordinate Attention, in CVPR, 2021.
[2] Zhang et al., ResNeSt: Split-Attention Networks, in arXiv 2020.

**Summary Of The Paper:**

This work proposes a previous knowledge channel attention module (PKCAM) that captures channel relations across the different layers to help enhance feature representation. The module can be integrated into the current ResNet series and show reasonable performance improvement over the baseline network on some benchmark datasets.

**Summary Of The Review:**

I think the introduction about the PKCAM module is very unclear. The experiments are not sufficient and convincing.

---

### Decision · Program_Chairs · 2022-01-20

**Decision:**

Reject

**Comment:**

This paper computes channel attention by considering feature maps across different layers, and named it the previous knowledge channel attention module (PKCAM). The reviewers find the proposed idea too straightforward and naive. Lack of technical contribution is one of the major criticisms. There are also correctness concerns with the submission. The authors have not provided any rebuttal.

We recommend rejecting the paper.